# Effect of Drying Methods on Lutein Content and Recovery by Supercritical Extraction from the Microalga *Muriellopsis* sp. (MCH35) Cultivated in the Arid North of Chile

**DOI:** 10.3390/md18110528

**Published:** 2020-10-26

**Authors:** Mari Carmen Ruiz-Domínguez, Paola Marticorena, Claudia Sepúlveda, Francisca Salinas, Pedro Cerezal, Carlos Riquelme

**Affiliations:** 1Laboratorio de Microencapsulación de Compuestos Bioactivos (LAMICBA), Departamento de Ciencias de los Alimentos y Nutrición, Facultad de Ciencias de la Salud, Universidad de Antofagasta, Antofagasta 1240000, Chile; francisca.salinas@uantof.cl (F.S.); pedro.cerezal@uantof.cl (P.C.); 2Centro de Bioinnovación, Facultad de Ciencias del Mar y Recursos Biológicos, Universidad de Antofagasta, Antofagasta 1240000, Chile; marticorena.paola@gmail.com (P.M.); claudia.sepulveda@uantof.cl (C.S.); carlos.riquelme@uantof.cl (C.R.)

**Keywords:** microalgae, *Muriellopsis*, spray drying, freeze-drying, lutein, supercritical fluid extraction

## Abstract

In this study, we determined the effect of drying on extraction kinetics, yield, and lutein content and recovery of the microalga *Muriellopsis* sp. (MCH35) using the supercritical fluid extraction (SFE) process. The strain was cultivated in an open-raceways reactor in the presence of seawater culture media and arid outdoor conditions in the north of Chile. Spray-drying (SD) and freeze-drying (FD) techniques were used for dehydrating the microalgal biomass. Extraction experiments were performed by using Box-Behnken designs, and the parameters were studied: pressure (30–50 MPa), temperature (40–70 °C), and co-solvent (0–30% ethanol), with a CO_2_ flow rate of 3.62 g/min for 60 min. Spline linear model was applied in the central point of the experimental design to obtain an overall extraction curve and to reveal extraction kinetics involved in the SFE process. A significant increase in all variables was observed when the level of ethanol (15–30% *v/v*) was increased. However, temperature and pressure were non-significant parameters in the SFE process. The FD method showed an increase in lutein content and recovery by 0.3–2.5-fold more than the SD method. Overall, *Muriellopsis* sp. (MCH35) is a potential candidate for cost-effective lutein production, especially in desert areas and for different biotechnological applications.

## 1. Introduction

In the past few decades, demand for bioprospection of microorganisms isolated from harsh environmental conditions has been increased because of their diverse biotechnological applications. Among microorganisms, microalgae are the most diversified photosynthetic organisms with high adaptability to different environmental conditions [1]. They are mainly classified as *Cyanophyta* (cyanobacteria), *Rhodophyta* (red algae), *Chlorophyta* (green algae), and *Chromophyta* (brown algae) [2,3].

Microalgae are characterized as natural sources of bioactive molecules such as phycobiliproteins, polysaccharides, carotenoids, lipids, fatty acids, polyphenols, and vitamins [4,5]. These compounds exhibit health benefits such as antibacterial, antifungal, antioxidant, and anticancer activities that are essential in pharmacological, nutraceutical, food, and biotechnological development [5,6,7]. Carotenoids belong to the class of terpenoids and are derived from the 40-carbon polyene chain and xanthophylls as their oxygenated derivatives in the presence of –OH groups (e.g., lutein), oxi-groups (e.g., canthaxanthin), or both (e.g., astaxanthin) [8,9,10]. Pure lutein is an orange-yellow, crystalline, and lipophilic solid whose chemical name is β, ε-carotene-3,3′-diol (C_40_H_56_O_2_). It is beneficial to human health due to its potential to ameliorate cardiovascular diseases [11], various types of cancer [12], and age-related macular degeneration [13] because of its antioxidant potential. The species known to accumulate carotenoids efficiently are *Chlorella* sp., *Chlamydomonas* sp., *Dunaliella* sp., *Muriellopsis* sp., and *Haematococcus* sp.

In this study, we used *Muriellopsis* sp. (MCH35), having the potential of producing carotenoids, especially lutein, as the main oxygenated carotenoid. This strain was isolated from an arid region of the north of Chile, as described previously by Marticorena et al. [14]. This area is located in Antofagasta, a well-known region for its high solar radiation, an environmental factor that favors microalgal growth [15]. Several studies have reported strategies to enhance microalgal carotenoid production using unfavorable environmental conditions such as nutrient deficiency, intense irradiation, and salinity to photo-bioreactors design [16,17,18]. These factors affect not only photosynthesis and productivity of cell biomass, but also pathways and cellular metabolism, and thus alter the cell composition [16]. The isolation and selection of microalgae is a prerequisite for the successful industrial production of biomass and beneficial compounds. Two factors are important in their successful industrial production: lutein content and biomass productivity. Other factors, such as the presence of cell wall or content of other carotenoids, can also be considered. A high lutein content in microalgae is necessary to achieve cost-effective and adequate amounts of extraction [19].

After the harvesting process, microalgal biomass pretreatment and extraction of bioactive compounds are most important because of their effect on bioactive molecule recovery. A drying process is the most common pretreatment method that increases the shelf life of naturally occurring bioactive biomass [20,21]. A dehydration process represents an essential step for reducing microbial growth, avoiding oxidative reactions, and improving bioactive compound extraction, although they are expensive [22,23,24]. However, these methods can alter the stability of labile bioactive compounds because of thermal breakdown [25]. Therefore, techniques such as spray-drying (SD) or freeze-drying (FD) that can adjust drying temperature have been recommended by researchers [23]. For the extractions of bioactive compounds, such as carotenoid from microalgae, many conventional methods, including maceration or soxhlet extraction, are used [19]. However, these methods are extensive and require a relatively huge amount of solvents, and thus are expensive and less eco-friendly [26]. Therefore, the use of green extraction technologies is increasing for the extraction of bioactive compounds. Green extraction technologies enhance extraction time, recovery, selectivity, and mass transfer with decreasing consumption of solvents. Supercritical fluid extraction (SFE), pressurized liquid extraction, microwave-assisted extraction, ultrasound-assisted extraction, and high-pressure homogenization are some examples of green extraction technologies [26].

The aim of this work is to study the effect of two different methods of biomass drying on lutein content and recovery, extraction kinetics, and yield of *Muriellopsis* sp. extracted by using the SFE process. The microalga was cultured in an open-raceways reactor under adapted outdoor conditions in the arid north of Chile. After harvesting of cells, SD and FD methods were employed for removing water content from the microalgal biomass and for knowing an individual carotenoid benchmark profile by using conventional methods. In the case of SFE, the parameters such as temperature (40–70 °C), pressure (30–50 MPa), and percentage of co-solvent (0–30% ethanol) were studied on the basis of Box-Behnken designs. In a similar way, the Spline linear model was used to control parameters for optimal lutein recovery from *Muriellopsis* sp. (MCH35). Drying pretreatment and optimal lutein extraction parameters for the microalgal biomass were studied with a focus on using the strain for future biotechnological applications and minimizing production cost when grown on a pilot-scale in a harsh environment.

## 2. Results and Discussion

### 2.1. Growth Parameters and Carotenoid Profile of Muriellopsis sp. (MCH35)

Photoautotrophic cultivation is a growing condition wherein light is an energy source, and carbon dioxide is an inorganic carbon source used to form chemical energy by using photosynthesis [27]. To evaluate the growth performance and parameters of *Muriellopsis* sp. (MCH35), batch tests were performed by using UMA5 culture medium [28]. The specific growth rate (µ) was found to be 0.085 d^−1^, as calculated by using Equation (1),
µ = Ln (*C*)⁄(*C*i)/t(1)
and is in accordance with previously reported data on the green microalga *Nannochloropsis gaditana* with similar environmental conditions, such as sufficient light and nutrient availability [29]. Therefore, the UMA5 culture medium is suitable for the cultivation of *Muriellopsis* sp. (MCH35) as it contains an adequate amount of nutrients required for the microalga. Nitrogen concentration and nitrogen:phosphorus ratio are widely recognized as determining factors for the growth and composition of microalgae [30,31]. An exponential increase was observed from day 0 to day 12 in biomass concentration starting with 0.44 g/L and reaching to the final concentration of 1.34 g/L (Figure 1A).

Chew et al. [32] reported the *Dunaliella* sp. biomass concentration of 1.5 g/L (in a volume of 3.4 L) at the laboratory scale. For *Phaeodactylum* sp., the biomass concentration of 1.38 g/L in 5 L medium was observed, while for *Chlorella vulgaris,* it was in the range of 0.6 to 1.08 g/L·d (in a volume of 1.5 L). Similar values were obtained for the first two strains, and the difference in biomass productivity of *Muriellopsis* sp. and *Chlorella vulgaris* can be due to the optimized laboratory conditions and differences in sizes of the two strains. The volumetric productivity of *Muriellopsis* sp. culture reached 75.73 mg/L·d on day 12, and the productivity per area was 16.81 g/m^2^·d (Figure 1A,B). High productivity was reported in the *Nannochloropis gaditana* cultures with 400 mg/L·d and incident irradiance of 1.100 µE/m^2^·s [33]. Our experiments on *Muriellopsis* sp. were carried out with an incident irradiance of 1.400 µE/m^2^·s, under outdoor conditions. The difference in productivities can be due to the different cultivation modes used, as a batch mode can affect productivity compared with semi-continuous mode. During the cultivation, it is essential to determine the variation in chlorophyll fluorescence (Fv/Fm) by focusing on the suitability of culture conditions. This parameter represents a measure of the quantum yield of PSII (photosystem II) and identifies any damage to the protein complex caused by photo-inhibition [34]. Figure 1B shows the values of photosynthetic efficiency ranging from 0.58 to 0.7 during the cultivation. An acceptable level of physiological acclimatization exhibited by *Muriellopsis* sp. was confirmed in our study (Figure 1B and Figure 2). In optimal cultivation conditions, the productivity of *Muriellopsis* sp. was ~0.60 at a pH of 7.9–8.2 and temperature of 16.3–19.6 °C, when cultivated in a semi-continuous mode by using Arnon culture medium [35]. Del Campo et al. [36] determined the limiting growth conditions such as pH of 6–9 and the temperature of 33 °C that stimulated carotenogenesis in *Muriellopsis* sp.

The main carotenoids present in the SD and FD biomass are shown in Table 1. These data are supported by Figure 3, showing an HPLC chromatogram with a diode array detector from the FD microalga.

The high individual carotenoid content was observed in FD biomass, except for astaxanthin that was 1.5 times higher for the SD method than the FD method. A similar drying effect was reported by Ryckebosch et al. [37] in which they evaluated the effect of the SD and FD methods on fresh biomass and the storage stability of lipids and carotenoids in the diatom *Phaeodactylum tricornutum.* Their study showed better results on carotenoid contents in the fresh and FD biomass of algae than that in SD biomass at 48 h (denominated as short-term storage). Lutein is the main carotenoid profile that *Muriellopsis* sp. shows [19,36,38]. Our results on lutein content (particularly lyophilized cells) were in a similar range with that reported previously (in range of 4.0 to 6.0 mg/g dry weight) [39,40]. Better lutein content was obtained by using both the drying methods than that reported by Molino et al. [41], who studied lutein production in different microalgae in a comparative manner. Our results were in a similar range as that reported by Del Campo et al. [8] and had an advantage that our cultures were produced at a large scale using seawater. This is relevant when developing massive cultures in areas with scarce water resources, such as desert areas (Antofagasta Region).

We found other carotenoids such as zeaxanthin, violaxanthin, astaxanthin, and β-carotene (can be seen in Figure 3) in low levels in *Muriellopsis* sp., similar to those reported by Del Campo et al. [36], with a total concentration of 5.10 ± 0.53 and 6.15 ± 0.63 mg/g for the SD and FD processes, respectively. Despite their low levels in *Muriellopsis* sp., their presence can contribute to pharmacological, nutraceutical, food, and biotechnological applications [7,42]. Lutein and zeaxanthin reduce age-related macular degeneration [43], while β-carotene prevents cataracts, skin diseases, and other illnesses like cancer [42,44]. Moreover, violaxanthin is a potential anti-photoaging agent acting against ultraviolet-B radiation (UV-B; λ 280–315 nm) [45], and astaxanthin is considered super vitamin E [46] for its stronger antioxidant activity (500 times more effective than α-tocopherol), preventing arteriosclerosis, coronary heart disease, and ischemic brain development [42,46]. Therefore, although it would be beneficial to optimize the extraction technology of lutein as suggested by Di Caprio et al. [47], our results provide preliminary information on the carotenoid profile of *Muriellopsis* sp. (MCH35).

### 2.2. Effects of Drying Processes on the Extraction Yield of Muriellopsis sp. (MCH35) by SFE

The experimental extraction conditions and results of the Box-Behnken designs from two-mode dry biomass by the SFE process are given in Table 2. The range of extraction yield (Y) was 0.11–7.84% (*w/w*) for the SD biomass and 0.76–6.05% (*w/w*) for the FD biomass. Table 2 shows the effect of co-solvent on extraction yield at all the conditions. The low levels of extracts were obtained without the addition of the modifier, such as 0.11–0.44% for the SD and 0.76–2.07% for the FD biomass.

Appendix A shows the statistical data of both drying processes with a *p*-value in terms of the goodness of fit of the model. It can be seen that ethanol, as a co-solvent, was a significant variable in the extraction process for two drying methods of biomass. These data are reinforced with the mathematical equations obtained in the statistical section and are summarized for the significant factors and their interactions (ethanol). Equations (2) and (3) were developed for an approximate mathematical model to maximize the extraction yield from *Muriellopsis* sp. dehydrated using the SD (Equation (2)) and FD (Equation (3)) methods:Yield (%, *w/w*) = 0.88 + 0.49·*Ethanol* + 0.0001·T·*Ethanol* − 0.001·P·*Ethanol* + 0.0007·*Ethanol*^2^(2)
Yield (%, *w/w*) = 7.85 + 0.10·*Ethanol* + 0.002·T·*Ethanol* + 0.0001·P·*Ethanol* − 0.002·*Ethanol*^2^(3)
where Yield is extraction yield in %, *w/w*, Ethanol is co-solvent in % *v/v*, T is temperature in °C, and P is pressure in MPa. These regression equations mathematically approach a model to maximize the yield from the two biomass drying methods based on the experimental results obtained in this work. Figure 4 also shows the results of extraction yields obtained by using the Pareto chart and RSM (Response Surface Methodology) using conditions such as 40–70 °C temperature and 0–30% *v/v* ethanol (extractant) with the optimal pressure of 30 MPa (original ranging from 30–50 MPa) using the two modes of dry biomass by SFE. Figure 4A,B represent the extraction yield of *Muriellopsis* sp. SD biomass, for which the optimum value was 6.40% *w/w* at 57.14 °C and 30 MPa pressure with 30% *v/v* ethanol as a co-solvent. These data are similar to the experimental conditions shown in Table 2 (run 11) with a value of 7.84% (55 °C, 30 MPa, and 30% ethanol). Figure 4C,D shows the significant factors obtained by the Pareto charts and the extraction yield obtained by RSM from the FD biomass. Its optimum value was similar to that of SD biomass with 5.96% at 59.67 °C temperature and 30 MPa pressure with 30% *v/v* ethanol as a co-solvent. In this case, run 26 (see Table 2) was the experimental condition closest to the optimum value, with 6.05% *w/w*. In the SFE process, the highest extraction yield and recovery of bioactive compounds can be achieved by optimizing some critical parameters. Indeed, many studies have shown an ability to modulate CO_2_ polarity by using co-solvents such as ethanol, and thereby increasing the extraction yields [48,49]. Our results showed an increase in the extraction yields using maximum extractant volume (30% *v/v*), similar to other studies. However, no significant interaction was found between temperature and pressure for the extraction yield in both the drying methods. Conversely, other reports accentuate the effect of factors including pressure or temperature on the extraction yield. Mehariya et al. [50] evaluated the pressure factor for measuring lutein extraction by the SFE process from the green microalga *Scenedesmus almeriensis.* They reported an increase in its yield with an increase in pressure from 25 to 55 MPa at 50 °C and above 65 °C (CO_2_ flow rate of 7.24 g/min).

Generally, an improving trend in the yield was observed in the case of FD biomass. In spite of various alternatives for drying methods available, both SD and FD are the most commonly used drying methods for high-value products [51]. In particular, FD is the best method for water removal and for the conservation of biochemical composition, although it is the most expensive process [51,52]. However, the SD process is faster than other methods, but it can damage different thermolabile components [37]. Therefore, our results showed that the solid-state of water during FD could protect the primary structure of the microalgal biomass with minimal reduction in volume and improving its general extraction yield.

### 2.3. The Effect of Drying Methods on Lutein Recovery of Muriellopsis sp. (MCH35) by SFE

SFE experiments based on Box-Behnken design were also performed to evaluate lutein content and recovery from *Muriellopsis* sp. (MCH35) using both drying processes. The SFE study focused on lutein as the main carotenoid as other individual carotenoids were in low levels. The effects of drying methods on lutein recovery are shown in Table 2 and complemented with Appendix A (data shown in Appendix A). The solvent factor was the most significant in both drying processes. Indeed, the interaction was stronger in the SD process as a quadratic factor. Both drying processes showed almost similar lutein content, with slightly more lutein content in FD biomass. FD biomass showed the highest lutein content (60.69 mg/g extract) during run 18 at 40 °C temperature, 50 MPa pressure, and 15% ethanol, followed by run 27 at 55 °C temperature, 50 MPa pressure, and 30% ethanol (54.48 mg/g extract). However, SD biomass showed lutein content of 47.14 mg/g extract (run 14) under 55 °C, 40 MPa pressure, and 15%, followed by run 1 (45.47 mg/g extract) at 40 °C temperature, 30 MPa pressure, and 15% ethanol. High pressure caused an increase in lutein content in FD biomass, contrary to SD biomass. In both processes, the temperature employed was between 40 and 55 °C and the amount of ethanol ranged from 15% to 30%, *v/v*. It is evident that the main difference in the lutein yield was due to the drying methods. Particularly, high pressure causes a major variation in solvent density so that it can have a good extraction yield for lutein. Moreover, the amount of free water present in *Muriellopsis* sp. (MCH35) can affect lutein extraction because of the difference in the degree of dehydration between SD and FD methods [52].

In accordance with the above results, lutein recovery obtained by using SD and FD methods is given in Table 2. Figure 5 shows the results obtained by Pareto charts and RSM curves for the combined effects of temperature (40–70 °C), pressure (30–50 MPa), and ethanol as a co-solvent (0–30% *v/v*) on lutein recovery from SD (Figure 5A,B) and FD biomass (Figure 5C,D). The presence of ethanol was significant in both methods. A similar trend was observed in the SD method because, again, the quadratic ethanol factor was relevant in lutein recovery (can be confirmed by Appendix A). The mathematical models are included in Equations (4) and (5) for the optimization of lutein recovery from SD and FD microalga, respectively. The equations summarize factors that were significant (ethanol as co-solvent), including their interactions.
Lutein recovery (%, *w/w*) = 32.06 + 2.48·*Ethanol* + 0.0002·T·Ethanol − 0.002·P·*Ethanol* − 0.04·*Ethanol*^2^(4)
Lutein recovery (%, *w/w*) = 8.06 + 0.43·*Ethanol* + 0.019·T·*Ethanol* + 0.002·P·*Ethanol* − 0.03· *Ethanol*^2^(5)
where Lutein recovery is measured as %, *w/w*, Ethanol is co-solvent in % *v/v*, T is temperature in °C, and P is pressure in MPa. These regression equations again approach a mathematical model obtained by the statistical software to maximize lutein recovery from both drying methods based on the experimental results. The highest lutein recovery obtained in this study was 49.84–74.52% *w/w* for the FD biomass in the presence of ethanol (15–30%, *v/v*). Conversely, the pressure and temperature factors were variable and hence were non-significant in the process. Moreover, this set of experimental factors also obtained the highest extraction yield. The optimal factors determined by the statistical analysis for the FD biomass were similar to those cited above. However, optimal lutein recovery was lower than the results mentioned above for the FD biomass, such as 60.47% *w/w* at 60 °C, 50 MPa, and 29.9% ethanol (*v/v*), approximately. On the other hand, the SD method achieved poor lutein recovery compared to that obtained by the lyophilization method. Thus, we obtained the best experimental result between 23.25% and 30.23% *w/w* in the presence of ethanol and at intermediate to high temperatures. A low-pressure trend was defined by the SD biomass in the presence of other optimal conditions (58.14 °C, 30 MPa, and 25.5% ethanol), with a value of 28.24% lutein recovery (*w/w*).

Other groups of scientists have also studied the effect of supercritical fluid parameters such as pressure, temperature, solvent, and CO_2_ flow rate on lutein recovery and purity from other sources. Mehariya et al. [50] investigated the effect of pressure (25–55 MPa), temperature (50 and 65 °C), and CO_2_ flow rate (7.24 and 14.48 g/min) on the green microalga *Scenedesmus almeriensis*. Their results showed improvements in the lutein recovery (~98%) and purity (~34%) with an increase in temperature, pressure, and CO_2_ flow rate. Yen et al. [53] reported 76.7% lutein recovery from *Scenedesmus* sp. in the presence of 70 °C, 40 MPa, and ethanol (30 mol%), which is in the similar range of our results. Wu et al. [54] extracted 87.0% of lutein from *Chlorella pyrenoidosa* by SFE in 4 h in the presence of 50 °C, 25 MPa, and modified CO_2_ with 50% ethanol. The factors optimized for lutein recovery from microalgae by the SFE process in our study are similar to those reported in previous studies. The effect of water content on the *Muriellopsis* biomass for lutein recovery has been evidenced in our SFE results. Some studies have reported that the optimal parameters for SFE can be obtained from samples that exhibit 3% to 12% of water content [55,56,57]. The presence of moisture in samples can act as a barrier for the diffusion of supercritical CO_2_ and extracted compounds [57]. However, other studies showed that the presence of water can enhance extraction kinetics and yields from plants and microalgae such as *Nannochloropsis oculata* [58,59], and water can play the role of co-solvent for polar compounds according to the type of matrix. We got the results emphasizing that optimal supercritical extraction was obtained in the samples with low moisture (lyophilized cells). Although FD is an expensive process for dehydration, an increase of lutein recuperation was significant. This can coalesce with growing the endogenous *Muriellopsis* sp. in seawater and under arid outdoor conditions to counteract the high costs of the dehydration process. Therefore, *Muriellopsis* sp. (MCH35) can be used as an efficient lutein producer for biotechnological applications, especially in desert areas.

### 2.4. Global Yield and Kinetic Curve of Muriellopsis sp. (MCH35)

Based on the yield results obtained for SFE by using Box-Behnken designs from the two modes of dry biomass of *Muriellopsis* sp., the condition selected for kinetic study was the central point of the experimental design, that is, 55 °C, 40 MPa, and CO_2_ + ethanol (85:15 *v/v*) flow rate, since temperature and pressure were non-significant (*p* < 0.05).

Figure 6A,B show overall extraction curves (OEC) of SD and FD biomass. A recovery of 2.54% and 5.10% of extracts was obtained after 150 and 160 min of extraction for SD and FD biomass, respectively. Similar weights (~2.0 g) of dry biomass were used for extraction, indicating that a double amount of extract was obtained by FD compared with that of SD. This gives an advantage to the biomass obtained by FD over SD since the resulting final powder is better in quality and quantity. This is due to the fact that drying by sublimation, as in the case of the FD technique, is better.

The OEC plotted for SD and FD biomass followed the SFE kinetics that were established by Meireles [60] and Jesus et al. [61]. The extraction process began with the CER period, characterized by the removal of easily extractable compounds by solvent and co-solvent, which was mainly controlled by the convective mass transfer in the fluid film around the powder particles. Following the CER period, the transition period began with a reduced extraction rate, wherein the extraction rate was controlled by mass transfer mechanisms through both convection and diffusion. This period is commonly called the FER period. When easily accessible solute became scarce in the microalgae matrix, intra-particle diffusion became the main mass transfer mechanism during SFE, and hence the OEC assumed a typical shape of diffusion curve with reduced extraction rate. 

From the fitted data by the Spline linear model in Figure 6A,B, the OEC parameters were estimated as given in Table 3. The calculated t_CER_ were 12.54 and 11.24 min with the accumulated extracts of 1.66% and 3.25%, and the recovery of 65.85% and 64.72% for SD and FD biomass, respectively. This is in agreement with a previous study that reported the recovery between 50% and 90% in the CER period [62]. Although t_CER_ of SD differs by less than 1 min and about 1.13% of the recovery compared to FD, the accumulated extract in FD was double that in SD. The calculated t_FER_ was 29.01 and 44.11 min for the accumulated extract of 2.01% and 4.15%, and the total recovery of 79.63% and 82.70% for SD and FD biomass, respectively. At this stage, the difference between t_FER_ of SD and FD was ~15 min, and the accumulated extract of FD was more than double that for SD. In our research, a recovery > 75% was achieved in the FER period. The M_CER_ and M_FER_ values represent the extraction rate of the CER and FER periods respectively [61], with values of 0.0026 and 4.6 × 10^−4^ g/min for SD, and 0.0057 and 5.3 × 10^−4^ g/min for FD. Therefore, the extraction rate that produced CO_2_ + Ethanol flow in the FD biomass was more than double that for SD. These values of the extraction rates were lower than the values reported for M_CER_ in peach almond oil (0.0084–0.0752 g/min) [63] and M_CER_ and M_FER_ of chañar almond oil (0.066–0.0124 g/min) [64].

On the other hand, Y values (mg extract/g biomass) in the CER and FER periods of FD showed a similar trend as above, since they were 1.9 and 2.5 times higher than the SD values. Y value (g extract/g CO_2_ + ethanol 85:15 *v/v*) represents the extract ratio in the supercritical phase at the bed outlet and were 2.3 and 1.3 times higher for the CER and FER periods respectively, in the FD biomass than in SD.

In the final stage of the process or DC period, the difference between the FD and SD biomass was observed. Table 3 shows the coefficients of determination (R^2^) for all periods, such as CER, FER, and DC obtained by means of the Spline linear model and their corresponding coefficients, such as b_o_, a_1_, a_2_, and a_3_. The CO_2_ + ethanol (85:15 *v/v*) flow rate was 3.305 g/min and solvent to feed ratio (S/F) of 8.2 to 247.1 (5–150 min) and 8.2 to 260.5 (5–160 min) for SD and FD, respectively. In different studies performed by using the SFE-CO_2_ technique, Sanzo et al. [65] reported the lutein recovery of ~47% from the *Haematococcus pluvialis* dry biomass with the CO_2_ flow rate of 3.62 g/min, 50 °C temperature, and 40 MPa pressure in 120 min. Yen et al. [53] reported the lutein recovery of 76.65% from the *Scenedesmus* sp. dry biomass with the CO_2_ flow rate of 1.45 g/min, 30% ethanol, 47.5 °C temperature, and 40 MPa pressure in 60 min.

The CO_2_ + ethanol mixture was efficient for the solubilization of lutein from *Muriellopsis* sp. (MCH35) under the established conditions studied by us and also supported by findings in other studies on *Haematococcus pluvialis* [65], *Scenedesmus* sp. [53], and *Scenedesmus almeriensis* [50]. Meireles [60] recommended to extend the SFE process up to the end of CER period, where the extraction rate is the highest, and sometimes it is necessary to extend the extraction process more than the CER period to attain the lowest production cost depending on the characteristics of the product to be extracted [66]. In the present study, the process period was up to 60 min to ensure the complete recovery of the extract of more than 85%.

## 3. Material and Methods

### 3.1. Microalgal Strain and Chemicals

The microalga *Muriellopsis* sp. (MCH35) was selected for this research and was isolated from freshwater in the arid north of Chile (Antofagasta Region), as previously described by Marticorena et al. [14]. This strain was deposited in the Spanish algae bank with accession number BEA_IDA_0063B. The UMA5 culture medium was of analytical grade and compounds were purchased from Merck (Darmstadt, Germany). The chemicals used for SFE were carbon dioxide (99% purity), purchased from Indura Group Air Products (Santiago, Chile), and ethanol co-solvent (99.5%), from Merck (Darmstadt, Germany). Other chemicals such as ultrapure water, ethanol, methanol, hexane, and acetone were of chromatographic grade (Sigma-Aldrich, Santiago, Chile) for the HPLC (Jasco Inc, Tokyo, Japan) system. Individual carotenoid standards such as lutein, zeaxanthin, violaxanthin, astaxanthin, and β-carotene were also procured from Sigma-Aldrich (Santiago, Chile).

### 3.2. Microalgal Culture Conditions

*Muriellopsis* sp. (MCH35) was maintained under controlled conditions in 20 L bottles at 20 ± 2 °C, under constant illumination at 80 µE/m^2^·s provided by fluorescent lamps, with constant aeration of 0.1 *v/v*/min without CO_2_ supply. The culture medium UMA5 was adapted to natural seawater conditions as described by Riveros et al. [28] and Marticorena et al. [14]. The inoculum was sub-cultured during the exponential growth phase on every 12th day by taking 10% of the old culture and 90% of fresh culture medium and were scaled up to open-raceway ponds of the surface of 36 m^2^ and capacity of 5.4 m^3^. Subsequently, *Muriellopsis* sp. (MCH35) cells were adapted to outdoor conditions under natural illumination with incident irradiance being evaluated by the light availability present at the installations of the Universidad Antofagasta (Antofagasta, Chile), where the reactor was located. Subsequently, the culture was maintained as a batch mode for 12 days with controlled pH by using the automatic injection of CO_2_. Finally, *Muriellopsis* sp. (MCH35) was harvested during its exponential growth phase by using a batch centrifuge (GEA separator, AS-1936076 model, Oede, Westphalia, Germany) at a flow rate of 2 m^3^/h and a maximum pressure of 0.3 MPa.

### 3.3. Growth Measurements

Dry biomass concentration (C_b_) was measured using 50 mL of the culture sample. The samples were passed through fiberglass filters (Ø1.6 µ Munktell Filter, Falun, Sweden) and were washed with distilled water. They were then dried in an oven at 105 ± 2 °C for 2 h (in triplicate) until the weight was stabilized. Biomass concentration was determined gravimetrically. In addition, biomass productivity was calculated in volumetric terms (Pb). Batch mode Equation (6) is mentioned below:*Pb* = (*Cf* − *C*i)⁄(tf-ti),(6)
where C is the biomass concentration in g/L and t is the time in days. The subscripts *i* and *f* denote initial and final measurements, respectively. On the other hand, an equation for calculating the specific growth rate (µ) of the culture was mentioned in the text as Equation (1).

In this equation, µ is the specific growth rate, C_i_ is the initial biomass concentration, and C is the biomass concentration at any time t during the exponential growth phase. The photosynthetic performance (Fv/Fm) was measured to determine cell viability. The maximum quantum yield of photosynthetic efficiency of photosystem II was achieved in the case of samples that were previously adapted to darkness for 15 min. The AquaPen-C fluorometer (Photon Systems Instruments, Drásov, Czech Republic) was used for this experiment, as described previously by Riveros et al. [28].

### 3.4. Drying Treatment on Microalgal Biomass

The biomass was divided into two groups: SD and FD. In the first group, the SD process was performed by using a LPG-5 Speed centrifuge spray-dryer (Jiangsu, China). The operating conditions for drying were as follows: air temperature of 185 ± 5 °C, outlet air temperature of 80 ± 5 °C, and a flow rate of 4 L/h. In the second group, the FD process was performed by using a Labconco FreeZone 18 L Benchtop Dry System (Labconco, Kansas City, MO, USA) at a temperature of −48 ± 5 °C and pressure of 2 Pa. Finally, all samples were packed in vacuum sealing plastic bags and stored at 4 ± 2 °C in the dark until use.

### 3.5. Detecting Individual Carotenoids

Individual carotenoids were extracted by using 5 mg of two-mode dry biomass for conventional extraction (benchmark extraction) or 50 mg from supercritical fluids (SF) extracts. Saponification of samples was then performed, and a tricomponent solution was added in all the samples as described by Cerón-García et al. [67]. This tricomponent solution was composed of ethanol:hexane:water in a proportion of 77:17:6 *v/v/v* and contained 0–60% *w/w* potassium hydroxide [68]. The supernatant was transferred into an amber vial for chromatographic analysis. Subsequently, individual carotenoids were separated and identified by using the HPLC system (Jasco Inc, Tokyo, Japan). It was equipped with a quaternary pump (PU-2089 s Plus), diode array detector, and RP-18 column (Lichrosphere, 5 µm × 150 mm) by using a method described by Cerón-García et al. [67]. In the mobile phase, solvent A was water/methanol (2:8, *v/v*), solvent B was acetone/methanol (1:1, *v/v*), and the detection wavelength was 450 nm at 25 °C of column temperature. External standards (Sigma-Aldrich) and their corresponding calibration curves were used to identify and quantify individual carotenoids such as lutein, zeaxanthin, violaxanthin, astaxanthin, and β-carotene. It was performed in triplicate (*n* = 3).

### 3.6. Recovery of Individual Carotenoids

The effect of operating conditions on the extraction of individual carotenoids was expressed in terms of recovery that was calculated on the basis of the initial mass of each compound, as per Equation (7) given below:Recovery (%) = (*Wc*⁄*Wt*) × 100(7)
where W_C_ is the mass of the extracted compound in mg, and W_t_ is the theoretical mass of the compound extracted conventionally (mg). Total carotenoids were extracted by the conventional method described in Section 3.5 and defined as benchmark extraction in Table 1.

### 3.7. SFE

The extraction was also performed by using a Speed Helix supercritical extractor (Applied Separation, Allentown, PA, USA), which was designed by Ruiz-Domínguez et al. [69], and the extraction process is described in detail in Figure 7. For each extraction, 2 g of SD or FD biomass of *Muriellopsis* sp. (MCH35) was used. It was previously ground and sieved using a standard sieve of 35 mesh of the Tyler series (particle size ≤ 0.354 mm), along with polypropylene wool and glass beads (ϕ = 1 mm), which was then inserted into a 24 mL stainless-steel extraction cell. In all the cases, the CO_2_ flow rate of 3.62 g/min was maintained, and each extraction was performed for 60 min. Extraction conditions for the microalga were selected on the basis of preliminary kinetic studies performed on *Muriellopsis* sp. (MCH35) and were set for 150–160 min to ensure the complete removal of bioactive compounds. The resulting extracts were collected in vials under dark conditions. The residual ethanol was evaporated under an N_2_ gas stream avoiding the oxidation of biomolecules in the extracts by using Flexivap Work-Station (Model 109A YH-1, Glas-Col, Terre Haute, IN, USA) for calculating the extraction yield. Then, the dried extracts with N_2_ atmosphere were stored at −20 ± 2 °C and in the dark until further analysis (at maximum 2 h).

#### 3.7.1. Experimental Design

Two Box-Behnken designs were implemented in random run order, generating 15 experimental conditions for each biomass drying mode independently (30 runs in total, refer to Table 2). Considering both the designs, three factors were evaluated at 3 different experimental levels, such as temperature (40, 55, and 70 °C), pressure (30, 40, and 50 MPa), and percentage of ethanol as a co-solvent (0%, 15%, and 30% *v/v*). The effect of the factors on different response variables such as extraction yield (Y) and lutein content and recovery were determined in triplicate (*n* = 3). Other carotenoids were excluded from the study because of their low quantities.

#### 3.7.2. Mathematical Modeling of Overall Extraction Curve (OEC) and Spline Linear Model

For kinetic analysis, an OEC between the optimal extraction time versus accumulated extract and lutein recovery was plotted. Extraction kinetics was performed at the central point of the experimental design (40 MPa, 55 °C, 15% ethanol *v/v*, and flow rate Q_T_ = 2 L/min = 3.62 g CO_2_/min), as described by Gilbert-López et al. [70]. Each SD and FD biomass sample was collected at preselected intervals of 150 and 160 min, respectively. In this assay, the extraction yield (Y%) and lutein recovery (as the majority carotenoid in the profile) were calculated at each point of the curve from both the modes. This assay was performed in duplicate with 15–20 points per sample.

The OEC was fitted to a Spline linear model containing three straight lines Equation (8), as shown in Equations (9)–(11). An adjustment was performed by using PROC REG and PROC NLIN of SAS University Edition Software (https://www.sas.com/en_us/software/university-edition.html) Finally, the fitted data from Equation (7) were plotted by using a Microsoft Excel-2016 spreadsheet. Each fitted line represents the following extraction stages related to the mass transfer mechanism: constant extraction rate (CER) period, falling extraction rate (FER) period that represents the stage at which both convection and diffusion in the solid substratum control the process, and diffusion-controlled (DC) periods, as described by Meireles [60]. In the CER period, the mass transfer rate for the CER period (M_CER_), as well as the time corresponding to the interception of the two lines (t_CER_), was computed from the Spline linear model. A similar procedure was followed for the FER period, and finally, the DC stage was computed. The experimental data obtained from the OEC were fitted. The mass ratio of the solute in the supercritical phase at the equilibrium cell outlet (Y_CER_) was obtained by dividing M_CER_ by the mean solvent flow rate for the CER period. A similar procedure was employed by Salinas et al. [64] in the mathematical calculations for obtaining almond oil from the chañar fruit (*Geoffroea decorticans*) by SFE.
(8)$$y=m_{Ext}=\left(b_{o}-\overset{i=N}{\underset{i=1}{\sum }}C_{i}\;a_{i+1}\right)+\overset{i=N}{\underset{i=1}{\sum }}a_{i}t$$

For one straight line:(9)$$y=\;m_{Ext}=b_{o}+a_{1}t\;\;\mathrm{for}\;t\;\leq \;t_{\mathrm{CER}}$$

For two straight lines:(10)$$y=m_{Ext}=b_{o}-t_{\mathrm{CER}}a_{2}+\left(a_{1}+a_{2}\right)t\;\mathrm{for}\;t_{\mathrm{CER}}<t\leq t_{\mathrm{FER}}$$

For three straight lines:(11)$$y=m_{Ext}=b_{o}-t_{\mathrm{CER}}a_{2}-t_{\mathrm{FER}}a_{3}+\left(a_{1}+a_{2}+a_{3}\right)t\;\mathrm{for}\;t_{\mathrm{FER}}<t$$
where y = response variable = m_Ext_ is the mass of extract, a_i_ (i = 0, 1, 2, 3) = linear coefficients of lines, t = time (min), t_CER_ = CER time (min), and t_FER_ = FER time (min). C_i_ for i = 1, 2 are the intercepts of these lines (for instance, C_1_ is the intercept of the first and second lines, and C_2_ is the intercept of the second and third lines).

Using the adjusted parameters, y_CER_ and y_FER_ calculated from t_CER_ and t_FER_ were calculated. Recoveries were then calculated at each time according to Equation (12) given below:(12)$$Recovery\;\left(\%\right)=\frac{y_{t}}{y_{time\;final\;OEC}}\;\left(100\right)$$

### 3.8. Statistical Analysis

Experimental designs and data analysis were performed by response surface methodology (RSM) and using the Statgraphics Centurion XVI^®^ (StatPoint Technologies, Inc., Warrenton, VA, USA) software. The effects of the factors on response variables in the separation process were assessed by using the pure error and considering a confidence interval of 95% for all the variables. The effect of each factor on response variables and its statistical significance were analyzed by using analysis of variance (ANOVA) (included in the Appendix A) and the standardized Pareto chart. 

The response surfaces of the respective mathematical models were also obtained, and a *p*-value of ≤ 0.05 was considered significant. All measurements were performed in triplicate (*n* = 3). The mathematical relationship of the response with three factors, *X*_1_, *X*_2_, and *X*_3_, involved in the design was approximated by using quadratic polynomial Equation (13) (second degree):(13)$$Z=\beta_{0}+\beta_{1}X_{1}+\beta_{2}X_{2}+\beta_{3}X_{3}+\beta_{12}X_{1}X_{2}+\beta_{13}X_{1}X_{3}+\beta_{23}X_{2}X_{3}+\beta_{11}X_{1}^{2}+\beta_{22}X_{2}^{2}+\beta_{33}X_{3}^{2}$$
where Z = estimate response, β_0_ = constant, β_1_, β_2_, and β_3_ = linear coefficients, β_12_, β_13_, and β_23_ = interaction coefficients between the three factors, and β_11_, β_22_, and β_33_ = quadratic coefficients. The multiple regression analysis was performed to obtain coefficients and equations that can be used to predict the responses.

## 4. Conclusions

The effect of drying methods as a pretreatment for *Muriellopsis* sp. (MCH35) biomass was studied for the optimization of lutein recovery extracted by the SF process. The production of microalga isolated from the arid north Chile was undertaken in seawater medium (UMA5) and arid outdoor conditions, with the focus on reduction of the operational costs. The production conditions were suitable especially for desert areas where solar radiation is high and fresh water is limited. The strain showed high biomass content and volumetric productivity values between 1.34 g/L and 75.73 mg/L·d respectively, after 12 days. Moreover, the *Muriellopsis* cells showed an average photosynthetic efficiency of 0.65, confirming the microalga was able to adapt to harsh environmental conditions. Moreover, the lutein content was in a similar range with that reported by other lutein-producing microalgae in outdoor conditions. The supercritical experimental outcomes showed that the modifier (ethanol) played a crucial role in the extraction in terms of yield, and lutein content and recovery. However, the parameters such as extraction temperature and pressure were non-significant in the extraction. The maximum optimal yield was similar under both the drying methods at the temperature of the medium to high, low pressure, and maximum extractant (30% *v/v* ethanol). The maximum lutein recovery of 74% (*w/w*) for the FD biomass was 2.5-fold higher than that for the SD biomass. Therefore, FD was the optimal pre-treatment to enhance the high-valuable extracts of *Muriellopsis* sp. (MCH35) as microalga adaptable to hostile growth environments for biotechnological applications.

## Figures and Tables

**Figure 1 marinedrugs-18-00528-f001:**
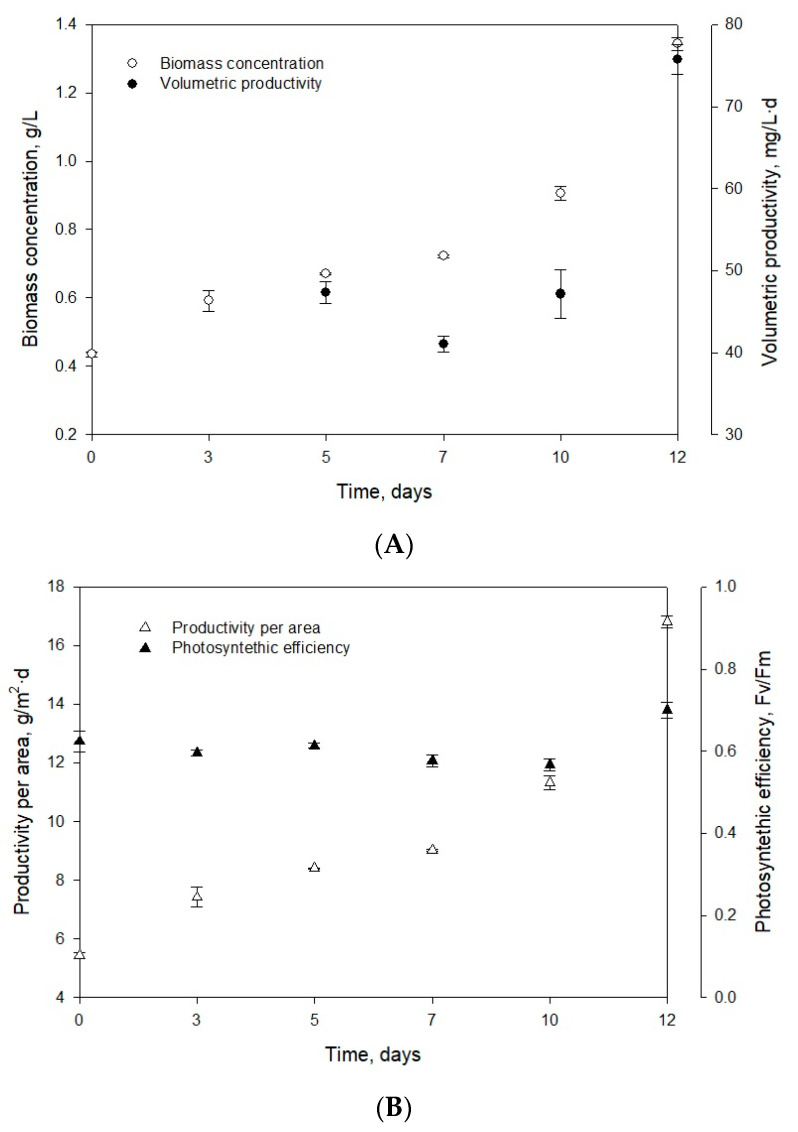
Evolution of (**A**) biomass concentration and volumetric productivity, and (**B**) productivity per area and photosynthetic efficiency of the strain *Muriellopsis* sp. (MCH35) in batch mode culture adapted under outdoor conditions in the arid north of Chile.

**Figure 2 marinedrugs-18-00528-f002:**
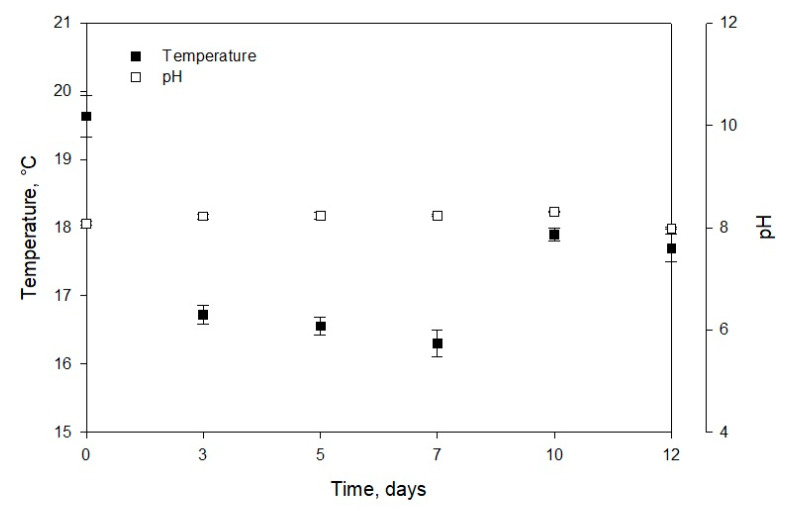
Temperature and pH and evolution of *Muriellopsis* sp. (MCH35) in a batch mode culture adapted under outdoor conditions in the arid north of Chile.

**Figure 3 marinedrugs-18-00528-f003:**
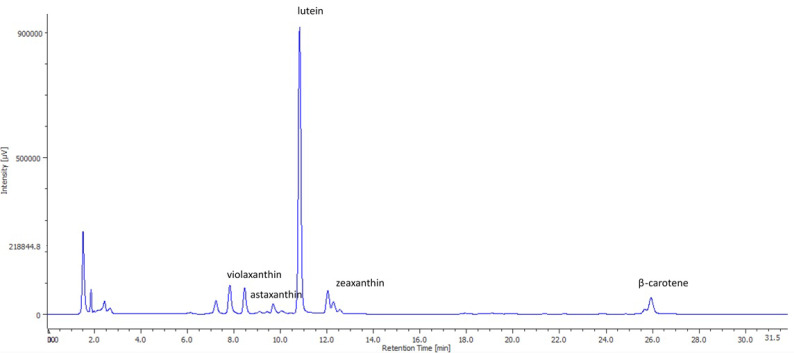
HPLC (High Performance Liquid Chromatography) chromatogram of individual carotenoids present in *Muriellopsis* sp. (MCH35), extracted under conventional extraction (freeze-dried (FD) biomass) and measured at 450 nm by a diode array detector.

**Figure 4 marinedrugs-18-00528-f004:**
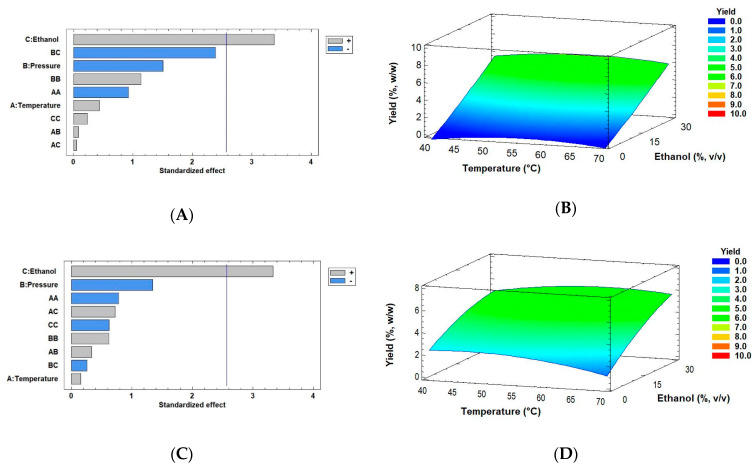
Pareto Charts and response surface curves of the combined effects of temperature (40–70 °C), pressure (30–50 MPa), and ethanol as co-solvent (0–30% *v/v*) on extraction yield from (**A**,**B**) spray- and (**C**,**D**) freeze-dried biomass of *Muriellopsis* sp. (MCH35), respectively. The ± signs are interpreted in the Pareto graph according to the area of significance of factor or interaction in the experimental design and Response surface curves were drawn at 30 MPa as optimal pressure.

**Figure 5 marinedrugs-18-00528-f005:**
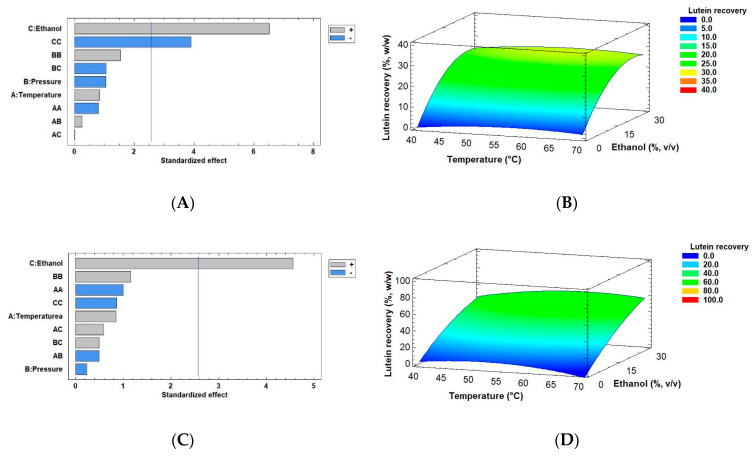
Pareto Charts and response surface curves of the combined effects of temperature (40–70 °C), pressure (30–50 MPa), and ethanol as a co-solvent (0–30% *v/v*) on lutein recovery from (**A**,**B**) spray- and (**C**,**D**) freeze-dried biomass of *Muriellopsis* sp. (MCH35), respectively. The ± signs are interpreted in the Pareto graph according to the area of significance of factor or interaction in the experimental design and Response surface curves were made at 30 MPa for spray-dried and 50 MPa for freeze-dried microalga as an optimal pressure.

**Figure 6 marinedrugs-18-00528-f006:**
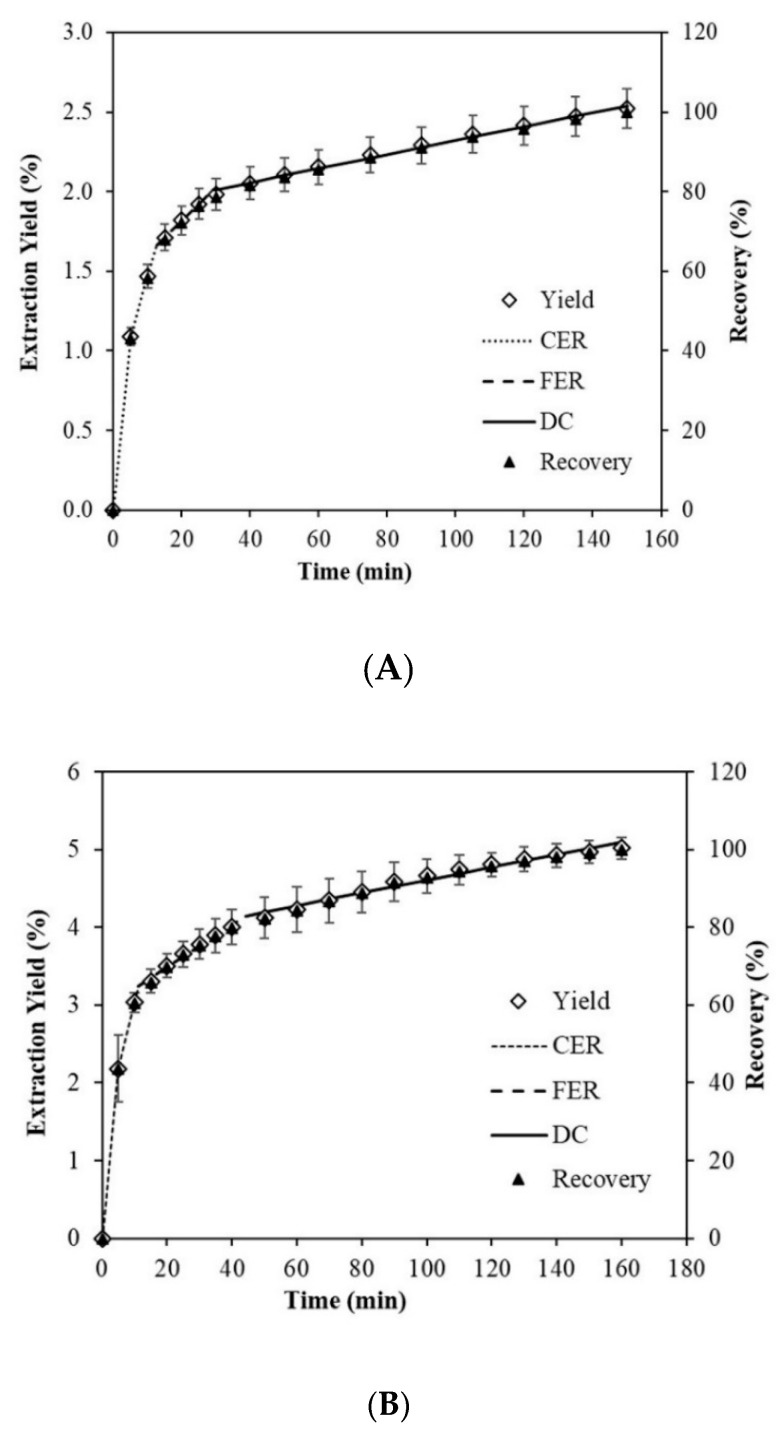
Overall extraction curves (OECs) of the experimental (◊) and predicted data for Spline (···· CER, ---- FER, — DC) models, and recovery (▲), at P = 40 MPa, T = 55 °C, CO_2_ + ethanol (85:15 *v/v*) flow rate (3.305 g/min), for Spray-Drying biomass (**A**), and Freeze-Drying biomass (**B**). Abbreviations: constant extraction rate period (CER); falling extraction rate period (FER); diffusion-controlled rate period (DC); Pressure (P) and Temperature (T).

**Figure 7 marinedrugs-18-00528-f007:**
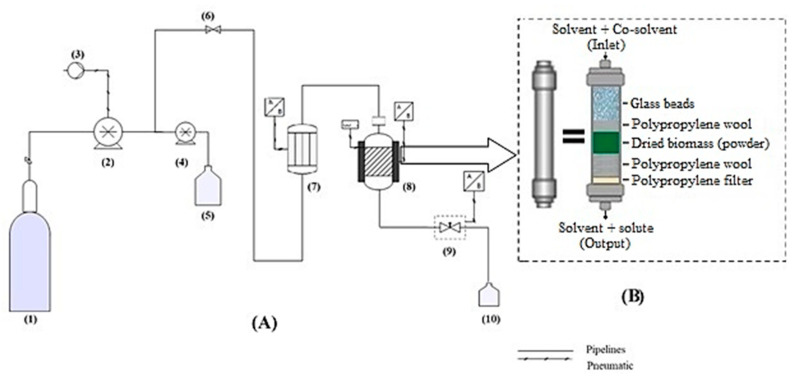
(**A**) Diagram of the SFE equipment (Applied Separations, Spe-ed, Allentown, PA). (1) CO_2_ cylinder, (2) CO_2_ pump, (3) Compressor, (4) Modifier pump, (5) Solvent tank, (6) Inlet valve, (7) Heater, (8) Extraction vessel and oven vessel, (9) Micrometric valve, (10) Sample collection. (**B**) Schematic diagram of the extraction vessel showing the way in which all elements are arranged.

**Table 1 marinedrugs-18-00528-t001:** Individual carotenoid profiles present in *Muriellopsis* sp. (MCH35) from two modes of dry biomass by the conventional method of extraction.

Biomass	Carotenoids Content (mg/g Biomass) *				
	Lutein	Zeaxanthin	Violaxanthin	Astaxanthin	β-carotene
**Spray-dried (SD)**	3.45 ± 0.20 ^b^	0.60 ± 0.04 ^b^	0.15 ± 0.10 ^a^	0.45 ± 0.15 ^a^	0.45 ± 0.04 ^b^
**Freeze-dried (FD)**	4.20 ± 0.30 ^a^	0.75 ± 0.06 ^a^	0.30 ± 0.20 ^a^	0.30 ± 0.02 ^b^	0.60 ± 0.05 ^a^

* Benchmark extraction. Mean values in the same column followed by different letters (a-b) are significantly different (*p* < 0.05).

**Table 2 marinedrugs-18-00528-t002:** Extraction yields (Y) and lutein content and recovery by SFE from spray- and freeze-dried *Muriellopsis* sp. (MCH35) using Box-Behnken experimental design. The general parameters were biomass loading = 2.0 g, CO_2_ flow rate = 3.62 g/min, and extraction time = 60 min.

**Biomass**	**Run**	**T**	**P**	**Ethanol**	**Yield**	**Lutein Content**	**Lutein Recovery**
		**(°C)**	**(MPa)**	**(%, *v/v*)**	**(%, *w/w*)**	**(mg/g Extract)**	**(%, *w/w*)**
**Spray-dried**	1	40	30	15	1.52 ± 0.08	45.47 ± 1.45	20.05 ± 0.48
	2	70	30	15	1.84 ± 0.09	43.95 ± 0.10	23.38 ± 0.13
	3	40	50	15	1.63 ± 0.08	34.27 ± 0.36	16.23 ± 0.04
	4	70	50	15	2.17 ± 0.02	34.70 ± 0.21	21.79 ± 0.92
	5	40	40	0	0.11 ± 0.01	2.60 ± 0.03	0.09 ± 0.3 × 10^−3^
	6	70	40	0	0.42 ± 0.02	6.28 ± 0.06	0.76 ± 1.2 × 10^−3^
	7	40	40	30	1.87 ± 0.09	27.69 ± 0.31	15.04 ± 0.05
	8	70	40	30	2.31 ± 0.08	23.69 ± 0.34	15.86 ± 0.14
	9	55	30	0	0.32 ± 0.10	1.27 ± 0.02	0.12 ± 0.03
	10	55	50	0	0.44 ± 0.02	7.80 ± 10^−3^	0.99 ± 0.01
	11	55	30	30	7.84 ± 0.27	13.30 ± 0.11	30.23 ± 0.45
	12	55	50	30	1.72 ± 0.06	43.76 ± 0.41	21.76 ± 0.30
	13	55	40	15	1.67 ± 0.03	31.21 ± 0.12	15.12 ± 0.52
	14	55	40	15	1.70 ± 0.09	47.14 ± 0.75	23.25 ± 0.18
	15	55	40	15	1.57 ± 0.03	38.68 ± 0.10	17.65 ± 0.63
**Biomass**	**Run**	**T**	**P**	**Ethanol**	**Yield**	**Lutein Content**	**Lutein Recovery**
		**(°C)**	**(MPa)**	**(%, *v/v*)**	**(%, *w/w*)**	**(mg/g Extract)**	**(%, *w/w*)**
**Freeze-dried**	16	40	30	15	4.79 ± 0.24	25.96 ± 1.18	29.62 ± 0.71
	17	70	30	15	4.55 ± 0.16	46.00 ± 0.26	49.84 ± 1.54
	18	40	50	15	1.62 ± 0.08	60.69 ± 2.06	23.39 ± 0.29
	19	70	50	15	2.32 ± 0.05	53.69 ± 2.41	29.64 ± 0.20
	20	40	40	0	2.07 ± 0.10	13.42 ± 0.31	6.62 ± 0.01
	21	70	40	0	1.15 ± 0.05	8.03 ± 0.39	2.20 ± 0.04
	22	40	40	30	2.69 ± 0.13	39.82 ± 1.29	25.52 ± 0.28
	23	70	40	30	3.77 ± 0.13	42.25 ± 2.55	37.90 ± 0.91
	24	55	30	0	0.76 ± 0.03	4.72 ± 0.30	0.86 ± 3.0 × 10^−3^
	25	55	50	0	1.18 ± 0.04	7.91 ± 1.00	2.22 ± 0.20
	26	55	30	30	6.05 ± 0.30	40.98 ± 4.49	59.04 ± 2.21
	27	55	50	30	5.74 ± 0.29	54.48 ± 0.83	74.52 ± 0.47
	28	55	40	15	3.91 ± 0.16	38.14 ± 1.82	35.55 ± 0.58
	29	55	40	15	3.33 ± 0.07	41.33 ± 0.33	32.72 ± 1.43
	30	55	40	15	3.07 ± 0.12	37.58 ± 1.09	27.47 ± 0.07

Acronyms: Temperature (T), Pressure (P), and Ethanol (co-solvent). Standard deviation was less than 5% in all operating conditions (SD ≤ 5%, *n* = 3).

**Table 3 marinedrugs-18-00528-t003:** Adjusted parameters of the spline linear model to SFE from *Muriellopsis* sp. biomass at 55 °C, 40 MPa, and CO_2_ + ethanol (85:15 *v/v*) flow rate.

Parameters	Stages of the General Extraction Curve					
	Spray-Drying			Freeze-Drying		
	CER	FER	DC	CER	FER	DC
Time (min.)	12.54	29.01	150.0	11.24	44.11	160.0
Accumulated extract (%)	1.66	2.01	2.54	3.25	4.15	5.10
Recovery (%)	65.85	13.78	20.37	64.72	17.98	17.30
Total Recovery (%)	65.85	79.63	100.0	64.72	82.70	100.0
M (g/min)	0.0026	4.6 × 10^−4^	9.1 × 10^−5^	0.0057	5.3 × 10^−4^	1.7 × 10^−4^
Y (mg extract/g biomass)	15.91	3.79	5.50	30.76	9.45	9.53
Y (g extract/g _(CO2 85%+ ethanol 15%)_	1.5 × 10^−3^	2.4 × 10^−4^	5.4 × 10^−5^	3.5 × 10^−3^	3.2 × 10^−4^	1.0 × 10^−4^
R^2^	0.9273	1.0000	1.0000	0.9451	1.0000	1.0000
**Drying Methods**	**b_o_**		**a_1_**	**a_2_**		**a_3_**
Spray-Drying	0.7098		0.0758	−0.0547		−0.0167
Freeze-Drying	1.3251		0.1711	−0.1437		−0.0192

$b_{0}$: Linear coefficient of the first line (CER); $a_{1},\;a_{2}\;\mathrm{and}\;a_{3}:$ Slopes of the lines 1, 2, and 3 corresponding to the periods CER, FER, and DC, respectively; $t_{CER},\;\mathrm{and}\;t_{FER}:$ Times in the intercepts of the lines 1 and 2 and the lines 2 and 3, respectively; $m_{EXT}\left(t\right):$ mass of the extract at time t in each period; $Y_{t}$: Variable response of the sixth, and the seventh row, for the considered stage (CER, FER, and DC).

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
