# Peer review of "Effect of Drying Methods on Lutein Content and Recovery by Supercritical Extraction from the Microalga Muriellopsis sp. (MCH35) Cultivated in the Arid North of Chile"

_marinedrugs, 2020, doi:10.3390/md18110528_

Round 1
Reviewer 1 Report
The paper of Ruiz Dominguez et al. reports on the effect of drying methods on lutein content and recovery by supercritical extraction. The authors investigate by statistical analysis the change in various parameters and tested the optimized ones.
The paper has not a great character of originality since the lutein content has been largely investigated in microalgae, so for SCF with different pressure, temperature and solvent conditions. In my opinion, the study should consider more specific parameters linked to the examined samples.
The authors should consider the effect of the addition of antioxidant as BHT the extraction solvent to prevent carotenoid oxidation. The authors should also consider the solvent/sample ratio, and the use of another solvent as methanol.
More what can happen to lutein that is not extracted? Is lutein degraded or oxidise? Which is the stability during a time of lutein extracted by these methods.
Why the Canthaxanthin is not observed?
Minor points
Line 99: please specify what is equation 6
Figure 1 and 2 analyse too many parameters and it is difficult to read results in the output. Maybe these figures can be re-organized and split in two or the panels.
Table 1. the error associated with lutein and violaxanthin has a decimal digit while the value two decimal digits. Please correct
Line 161: please associate an error to the reported values in mg/g
Line 189: please explain the meaning of terms in equations 1, 2, 3 and 4
Author Response
Dear Reviewer # 1
We want to thank you for your constructive comments. See below for our responses in the attached file and revised manuscript.
Best regards.
MC. Ruiz-Domínguez

Reviewer 2 Report
Title: Effect of drying methods on lutein content and recovery by supercritical extraction from the microalga Muriellopsis sp. (MCH35) cultivated in the arid north of Chile
The authors described the effect of drying on extraction kinetics, yield, and lutein content and recovery of the microalga Muriellopsis sp. (MCH35) using the supercritical fluid extraction (SFE) process. They used two drying techniques, i.e. spray-drying and freeze-drying for dehydrating the microalgal biomass, and the obtined results were compared. The general conclusion is that Muriellopsis sp. (MCH35) is an excellent candidate for lutein production, the adaptability to hostile growth environments being an additional advantage.
In my opinion, the study is well documented, written in a clear and explicit manner. I think that it can be published after minor corrections.
I have the following observations:
- L 98-99 “The specific growth rate was found to be 0.085 d–1 as calculated by using Equation 6” - for easier reading the equation should be provided here as Eq. 1, because it is found just at L 390, or the mention can be omitted here in text. By reading from the beginning, Eq. 6 is before Eq. 1.
- The images are not clear enough.
- The last phrase in “Abstract” and “Conclusions” is quite similar. The information from the abstract and the conclusions must be different, so a reformulation of these phrases would be preferable.
Author Response
Dear Reviewer # 2,
We want to thank you for your constructive comments. See below for our responses in the attached file and modified manuscript.
Best regards.
Dr. MC Ruiz-Domínguez

Round 2
Reviewer 1 Report
The author replied to my concerns highlighting the novelty of the paper. More, some useful details have been added in the revised version.